# Overcoming Underpowering in the Outcome Analysis of Repaired—Tetralogy of Fallot: A Multicenter Database from the CMR/CT Working Group of the Italian Pediatric Cardiology Society (SICPed)

**DOI:** 10.3390/diagnostics13203255

**Published:** 2023-10-19

**Authors:** Lamia Ait-Ali, Benedetta Leonardi, Annalisa Alaimo, Giovanna Baccano, Elena Bennati, Valentina Bucciarelli, Alberto Clemente, Silvia Favilli, Francesca Ferroni, Maria Cristina Inserra, Luigi Lovato, Antonella Maiorano, Simona Anna Marcora, Chiara Marrone, Nicola Martini, Gianluca Mirizzi, Giulia Pasqualin, Giuseppe Peritore, Giovanni Puppini, Camilla Sandrini, Francesca Raimondi, Francesco Secchi, Gaia Spaziani, Nicola Stagnaro, Stefano Salvadori, Aurelio Secinaro, Bertrand Tchana, Gianluca Trocchio, Davide Galetti, Federica Pieroni, Stefano Dalmiani, Francesco Bianco, Pierluigi Festa

**Affiliations:** 1Institute of Clinical Physiology, National Research Institute, 56123 Pisa, Italy; lamia.ait-ali@cnr.it (L.A.-A.); stefano.salvadori@cnr.it (S.S.); 2Pediatric Cardiology and GUCH Unit, Fondazione “G. Monasterio” CNR-Regione Toscana, 541200 Massa, Italy; marrone@ftgm.it (C.M.); nicola.martini@ftgm.it (N.M.); gigifesta@ftgm.it (P.F.); 3Department of Pediatric Cardiology, Cardiac Surgery and Heart Lung Transplantation, Bambino Gesù Children’s Hospital, IRCCS, 00100 Rome, Italy; benedetta.leonardi@opbg.net; 4U.O.C. Cardiologia Pediatrica, P.O. “G. Di Cristina”, ARNAS Civico, 90123 Palermo, Italy; annalisa.alaimo@gmail.com; 5Department of Pediatric Cardiology, Centro Cardiologico Pediatrico Mediterraneo, 98039 Taormina, Italy; giovanna.baccano67@gmail.com; 6Pediatric Cardiology, Azienda Ospedaliero-Universitaria Meyer, 50100 Florence, Italy; elena.bennati@meyer.it (E.B.); silvia.favilli@meyer.it (S.F.); francesca.raimondi@gmail.com (F.R.); g.spaziani@meyer.it (G.S.); 7Department of Pediatric and Congenital Cardiac Surgery and Cardiology, Azienda Ospedaliero-Universitaria Ospedali Riuniti Ancona “Umberto I, G. M. Lancisi, G. Salesi”, 60123 Ancona, Italy; valentina.bucciarelli@ospedaliriuniti.marche.it; 8Department of Radiology, Fondazione Toscana Gabriele Monasterio, 56123 Pisa, Italy; clemente@ftgm.it; 9Department of Radiology, Cardinal Massaia Hospital, 14100 Asti, Italy; 10Department of Radiology, University Hospital Vittorio Emanuele Catania, 95100 Catania, Italy; c.inserra@yahoo.it; 11Pediatric and Adult Cardiovascular, Thoraco-Abdominal and Emergency Radiology Unit, IRCCS Azienda Ospedaliero-Universitaria di Bologna, Via Massarenti 9, 40138 Bologna, Italy; luigi.lovato74@gmail.com; 12Cardiologia Pediatrica, Ospedale Pediatrico Giovanni XXIII di Bari, Via Amendola 207, 70100 Bari, Italy; an.maiorano@alice.it; 13USSD Cardiologia Pediatrica, ASST Grande Ospedale Metropolitano Niguarda, 20126 Milano, Italy; simonaanna.marcora@ospedaleniguarda.it; 14Division of Cardiovascular Medicine, Fondazione G. Monasterio CNR-Regione Toscana, 56123 Pisa, Italy; gianluca.mirizzi@ftgm.it; 15Department of Radiology, IRCCS Policlinico San Donato, San Donato Milanese, 20097 Milano, Italy; giulia.pasqualin@grupposandonato.it (G.P.); francesco.secchi@unimi.it (F.S.); 16U.O.C. di Radiodiagnostica, P.O. “G. Di Cristina”, ARNAS Civico, 90123 Palermo, Italy; giuseppe.peritore@hotmail.it; 17Department of Radiology, University of Verona, 37100 Verona, Italy; giovanni.puppini@aovr.veneto.it; 18Division of Cardiology, Department of Medicine, University of Verona, 37100 Verona, Italy; sandrini.camilla@gmail.com; 19Department of Cardiology and Cardiovascular Surgery, Papa Giovanni XXIII Hospital, 24100 Bergamo, Italy; 20Radiology Unit, IRCCS Istituto Giannina Gaslini, 16147 Genoa, Italy; nicola.stagnaro@gmail.com; 21Advanced Cardiothoracic Imaging Unit, Department of Imaging, Bambino Gesù Children’s Hospital, IRCCS, 00100 Rome, Italy; aurelio.secinaro@opbg.net; 22Pediatric Cardiology Unit, General and University Hospital, 43121 Parma, Italy; btchana@libero.it; 23Pediatric Cardiology Department, Giannina Gaslini Research Institute and Children Hospital, 16100 Genova, Italy; gianlucatrocchio@gaslini.org; 24Inf Department, Fondazione “G. Monasterio” CNR-Regione Toscana, 541200 Massa, Italy; davide.galletti@gmail.com (D.G.); federica.pieroni@ftgm.it (F.P.); stefano.dalmiani@ftgm.it (S.D.)

**Keywords:** tetralogy of Fallot, cardiac magnetic resonance imaging, congenital heart disease, Italian registry, repaired tetralogy of Fallot

## Abstract

**Background:** Managing repaired tetralogy of Fallot (TOF) patients is still challenging despite the fact that published studies identified prognostic clinical or imaging data with rather good negative predictive accuracy but weak positive predictive accuracy. Heterogeneity of the initial anatomy, the surgical approach, and the complexity of the mechanism leading to dilation and ventricular dysfunction explain the challenge of predicting the adverse event in this population. Therefore, risk stratification and management of this population remain poorly standardized. **Design:** The CMR/CT WG of the Italian Pediatric Cardiology Society set up a multicenter observational clinical database of repaired TOF evaluations. This registry will enroll patients retrospectively and prospectively assessed by CMR for clinical indication in many congenital heart diseases (CHD) Italian centers. Data collection in a dedicated platform will include surgical history, clinical data, imaging data, and adverse cardiac events at 6 years of follow-up. **Summary:** The multicenter repaired TOF clinical database will collect data on patients evaluated by CMR in many CHD centers in Italy. The registry has been set up to allow future research studies in this population to improve clinical/surgical management and risk stratification of this population.

## 1. Introduction

Tetralogy of Fallot (TOF) is the most frequent cyanotic congenital heart disease (CHD), with an incidence estimated at 1/3500 live births. The latter is represented by four distinctive features: a perimembranous ventricular septal defect (VSD), the aortic root that overrides the septum, and therefore also the VSD, accompanied by right ventricular (RV) hypertrophy, and right ventricular outflow tract (RVOT) obstruction; the latter derived from the malalignment and the anteriorly deviation of the conal septum. [1] Consequently, TOFs can be considered a cono-truncal malformation; in fact, due to this malalignment, the ventricular septum results anteriorly deviated, leading to an underdevelopment of the RV outlet portion of the RV, also known as infundibular, which results in an RV outflow that is narrowed and obstructed [1,2].

From the late 70s to nowadays, cardiac surgery has rapidly evolved and improved. This has permitted childhood intra-cardiac TOF corrections and has allowed the majority of the TOF patients (more than 90%) to survive to adulthood. This population of patients that underwent TOF cardiac surgery during childhood, live to date with a repaired TOF and may present with a wide range of residual anatomical/hemodynamic/thoracic residual issues, needing to be addressed in the future in their life. However, their optimal treatment is hampered by their heterogeneity due to significant differences in initial cardiac/pulmonary anatomy, surgical techniques of repair, as well as the era of them and subsequent surgical/interventional procedures along the follow-up [1,2].

The RVOT obstruction can occur at multiple levels (subvalvular, valvular, and supravalvular); therefore, it is very common that TOF may be accompanied by infundibular pulmonic stenosis and annular hypoplasia of the pulmonary valve, the main pulmonary artery, and branches, along with any peripheral pulmonary artery stenosis [2].

The surgical repair consists of the closure of the interventricular defect and relief of the obstruction of the right outflow tract, resecting the RV infundibular muscularity, and very often extending the incision to the pulmonary annulus, opening it utilizing a trans-annular patch. In more severe cases, pulmonary pathway-associated anomalies need to be addressed. All this, inevitably, causes severe pulmonary valve dysfunction that, together with an RV infundibulum negatively remodeled, leads to an RV overload [1,2,3,4].

In addition, ROVT aneurysmal dilatation, succeeding pulmonary regurgitation, anastomotic site stenosis, main/branches pulmonary arteries stenosis, and distortions, along with progressive aortic root dilatation and subsequent aortic regurgitation, biventricular failure, and residual VSD, are also possible as follow-up sequelae [1,2,3]. It is also estimated that in the long-term follow-up, 30% of patients with operated TF will have atrial arrhythmia and 10% will have complex ventricular arrhythmias (VT). In contrast, the risk of sudden death has been estimated at around 0.2%/year [2].

It is well known that the repair leads to right ventricular scarring and the extension of such myocardial fibrosis is documented on the cardiac magnetic resonance (CMR) by the presence of late gadolinium enhancement (LGE). Scar burden visualized by magnetic resonance imaging has been shown to be predictive for inducible VT in this population. In fact, myocardial fibrosis results in decreased myocardial contractility and relaxation, and serves as a potential substrate for scar-induced VT. Similarly, other measures such as myocardial strain and rotation patterns, location and extent of myocardial fibrosis, LV systolic circumferential strain rate, and other parameters derived from magnetic resonance all contribute to our understanding of higher-risk substrates for VT. Given that the method of performing the surgical correction has changed over time, from a transannular patch with a transventricular approach to that with a transatrial approach, the obvious consequence is that even the surgical era is also associated with risk of VT. Arrhythmias are the most important problem to be taken into consideration in TOF. It has been estimated that in the long-term follow-up of TOF patients, 30% of patients will have atrial arrhythmia and 10% will have complex ventricular arrhythmia, and the mortality risk triples after the third postoperative decade, with sudden cardiac death (SCD) being the most common cause of late mortality [2,3,4,5].

Pulmonary valve replacement (PVR) is recommended, at some moment of the patient’s life, as the best choice to resolve the RV overload due to pulmonary valve incompetence preventing the progressive exercise intolerance, arrhythmia due to the right heart failure, and sudden death. However, to this day, indications and optimal timing of PVR are unclear in the asymptomatic population, and the extent to which PVR really improves survival is still unknown since it does not completely eliminate right ventricular scarring. In addition, the bioprostheses valves used for PVR are destined to deteriorate in a short time, both for the structural valve failure and the increased risk for endocarditis [3,4,6].

Given all these premises, it is clear that many questions regarding the follow-up and late management, along with a risk stratification, of repaired TOF are still unanswered, with an imperative need to be addressed. Hence, the multicenter repaired TOF clinical database registry, that we here present, has been set up to allow future research studies in this population to improve clinical/surgical management and risk stratification of this population.

## 2. Materials and Methods

### 2.1. Trial Rationale

In clinical practice, the diagnosis and monitoring of any residual defects and adverse events in this population are based on clinical evaluation and ECG on non-invasive imaging diagnostic tools. Echocardiography remains the first-choice method; however, CMR is considered the method of choice for measuring volumes and functions (both atrial and mostly important ventricular) and evaluating pulmonary regurgitation and the distribution of flows in the pulmonary branches. Moreover, thanks to the ability to characterize the myocardium, CMR allows for evaluating both focal and diffuse myocardial fibrosis. Therefore, CMR has a pivotal role in the lifelong follow-up and decision-making process for the repaired TOF population [3].

A pulmonary valve of adequate size for the body surface, surgical or interventional implanted, is today the only therapeutic option to contrast progressive right ventricular dilatation. However, there is currently no valve or pulmonary conduit of lifelong duration. Therefore, multiple procedures are often necessary over a lifetime. In addition, there is no clear evidence that the pulmonary valve reduces arrhythmia or improves survival in this population [7].

Published studies identified prognostic clinical or imaging data with good negative predictive accuracy but weak positive predictive accuracy [3]. The increased risk of an adverse outcome was reported to be associated with progressive dilation and dysfunction of the left ventricle [8,9]. However, heterogeneity of the initial anatomy, the surgical approach, and the complexity of the mechanism leading to dilation and ventricular dysfunction explain difficulties in predicting the adverse event in this population [10,11,12].

Therefore, risk stratification and management of this population remain poorly standardized and debated. Also, there is a lack of knowledge of very long-term survival due to only 40–50 years of follow-up available for repaired TOF [13,14,15].

Multicenter clinical databases are helpful for the evaluation and the study of chronic diseases, reducing the heterogeneity of the population enrolled and improving the statistical empowerment [16].

The CMR/CT WG of the Italian Pediatric Cardiology Society clinical database of repaired TOF involves many pediatric cardiology centers managing and treating treatment CHD in Italy. Its aim is to collect a large cohort of repaired TOF anamnestic, clinical, imaging data, and adverse events. The study data will be included in further studies to contribute to identifying anamnestic and clinical-instrumental parameters predictive of adverse events in repaired TOF patients and, therefore, contribute to improving clinical and surgical management of this complex population.

### 2.2. Study Design

This observational study will include all eligible patients retrospectively and/or prospectively repaired TOF patients who were evaluated for a CMR study in participating in a CMR Lab from 1 January 2004 to 31 December 2024. Eighteen Italian CMR sites have accepted to participate in this study. Eight have already received the local ethical committee’s approval and started the data collection. The study has also been registered on ClinicalTrials.gov with the ID: NCT05288894. This study protocol also follows and adheres to the SPIRIT recommendations.

### 2.3. Inclusion Criteria

Patients with repaired TOF/pulmonary atresia + VSD, double outlet right ventricle (DORV);Age > 10 years, as CMR is indicated in younger patients only in exceptional cases.

### 2.4. Exclusion Criteria

Different associated complex pathology such as MAPCAs, atrioventricular canal or Ebstein;Incomplete CMR study;Patients who do not consent to the study;Contraindication to CMR.

### 2.5. Data Collection and Management

#### 2.5.1. The Software Platform for the Implementation of the Clinical Database

Data will be recorded into a web-based case report form that will be accessible through a restricted web-based site for the Fallot-CMR Multicenter Italian Database, developed as centralized support to the network of centers.

The multicenter repaired TOF clinical database is based on the Rare Diseases Platform (RDP), a software for clinical data management already used to develop and support other patient registries such as the Dravet syndrome Italian registry and European RESIDRAS Platform (REF); the Tuscany registry for development of mental and epileptic encephalopathies and medical comorbidities (REMEDIES) for the DECODE-EE project; and the celiac disease Tuscany registry.

RDP implements a web-based application, which allows different hospitals or research centers to manage a patient’s medical records. Medical records keep a patient’s clinical history, where some data can be entered once, and some are serials. As a rule, the investigator can access only the information associated with the medical records created and maintained within the hospitals.

However, to support the cooperation among centers, the information can be shared among hospitals with different approaches, i.e., one possible technique consists of requesting a medical record that is subsequently handed over to another hospital. Data extraction in tabular form is available for any managed entity where required data is pseudonymized to mask fields that could lead to the disclosure of personal data belonging to a single patient.

For the multicenter repaired TOF clinical database, the most relevant anamnestic, clinical, and imaging data of repaired TOF were identified.

The platform is designed to manage health data; hence, particular attention is paid to personal data protection. These requirements are addressed by the General Data Protection Regulation (GDPR), and the data are managed and stored according to the highest standards in data management.

Patients will be assigned a unique identification code by which they will be identified. The enrollment site will give the identification number after a patient consents to participate in the registry. The link between the patient and the associated identification number will remain at the enrolling center under restricted access.

Each participating center will be responsible for collecting demographic, anamnestic, clinical, and image data sets defined by this study. In addition, any adverse events in the follow-up will be organized.

#### 2.5.2. Data Collection

The list of variables that will be collected is itemized in Figure 1.

In summary, for each patient, the following will be collected:*Demographic information form;**Baseline information about the patient’s health at the time of enrollment;**Anthropometric data;**Pre-operative anatomy* data: native anatomy impacts the surgical and interventional history of repaired TOF, therefore, when available, the data about the annulus and pulmonary branches anatomy, as well as associated extracardiac defects, will be collected (Figure 2) [17].

*Surgical history*: type and age at intracardiac repair, and previous palliation will also be reported as they influenced the clinical follow-up of this population [2,18].*Electrocardiogram* (ECG) data: QRS length and QRS fragmentation were both identified as risk factors for adverse events in repaired TOF. Moreover, data from Holter monitoring, when available as significant atrial or ventricular arrhythmias, will also be reported (Figure 3) [2,19,20,21].*Transthoracic echocardiography* (TTE) data: TTE is the first diagnostic tool in this population, allowing the evaluation of many of the anatomic and hemodynamic abnormalities in this population [22]. In particular, the current clinical routine is in tricuspid regurgitation and estimated Doppler systolic right ventricular pressure. The main parameters evaluated by TTE, tricuspid annular peak systolic velocity (TAPSE), and myocardial acceleration during isovolumic contraction have also been investigated in evaluating RV function in this population [2,18]. Therefore, data on bi-ventricular and atrial dimensions and function according are included in the dataset of the study.*Cardiac Magnetic Resonance* (CMR) data: CMR is the gold standard. CMR is considered the reference standard for quantifying RV size, function, and PR in patients with repaired TOF [14,15]. Biventricular volumes and function are predictors of adverse outcomes in repaired TOF and atrial volumes, and function also emerged as prognostic predictors. The LGE score also has been demonstrated to be associated with ventricular arrhythmias, and there is an increasing interest in the prognostic role of T1 mapping in this population. Moreover, CMR allows for the evaluation of the anatomy and flow of the main and pulmonary branches, and it is the unique modality that is able to quantify the pulmonary regurgitation and the pulmonary flow distribution. Therefore, all the anatomic and functional parameters evaluated by CMR comprehensively, as well as the evaluation of the aortic valve for regurgitation and measurement of aortic size, are included in the study dataset (Figure 4).*Cardiopulmonary test* (CPET) data: CPET is routinely used in the function evaluation in repaired TOF and provides prognostic predictors in this population. Therefore, any available date, Vo2 mL/kg/min, predicted, Vo2/HR, as well as the type of the protocol used, are included [21,22] (Figure 5).*Cardiac catheterization* (CC) data: CC is mainly restricted in repaired TOF for interventional procedures and when non-invasive evaluation is inconclusive. In case of any cardiac catheterization, the date about atrial and ventricular pressure can be recorded [20,22].*Cardiac tomography* (CT) data: even if a CT scan is rarely indicated in the routine follow-up, in selected cases with contraindication to CMR or for the evaluation of ferromagnetic device, a CT scan could be useful for the evaluation of coronary arteries, conduit calcification. A CT scan may also be considered as an alternative for ventricular quantification in patients unable to undergo CMR [19]. CT data of bi-ventricular and bi-atrial volumes and bi-ventricular function, as well as diameters of great vessels, could be reported.*Brain natriuretic peptides* (Nt-ProBNP and BNP) are prognostic biomarkers in this population and will be also recorded [22,23].

**Figure 3 diagnostics-13-03255-f003:**
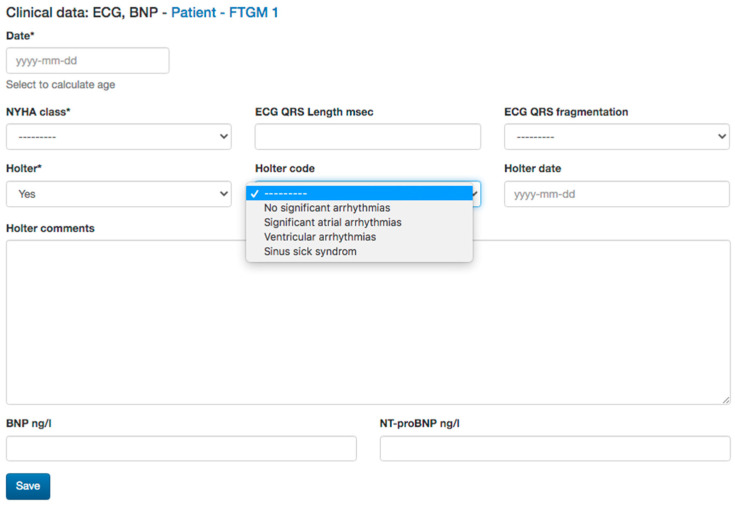
Functional class, ECG, and natriuretic peptides data collected.

**Figure 4 diagnostics-13-03255-f004:**
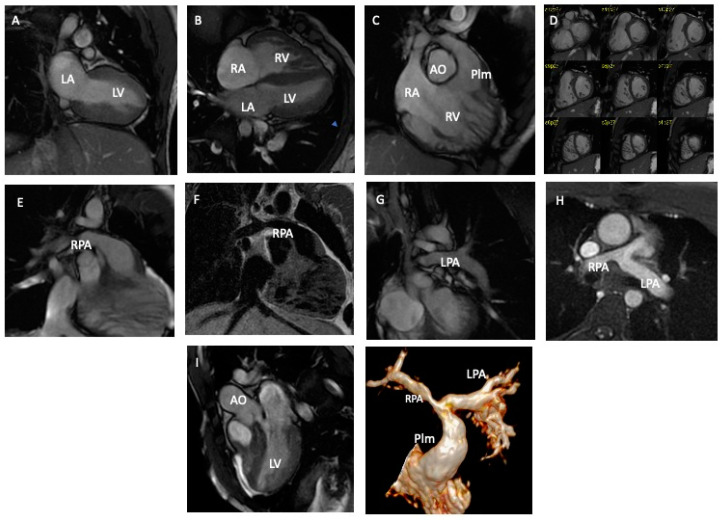
Cardiac magnetic resonance (CMR) imaging evaluation of a young man with tetralogy of Fallot operated utilizing a transannular patch (TAP) (Panels **A**–**C**,**I**). The RV is mildly dilatated (120 mL/m^2^) (Panels **D**,**E**). There is a RPA stenosis with a mild hypoperfusion for the right lung (Panles **F**–**H**). RV: right ventricle. LV: left ventricle. RA: right atrium. LV: left atrium. AO: aorta. Plm: pulmonary artery. RPA: right pulmonary artery. LPA: left pulmonary artery.

**Figure 5 diagnostics-13-03255-f005:**
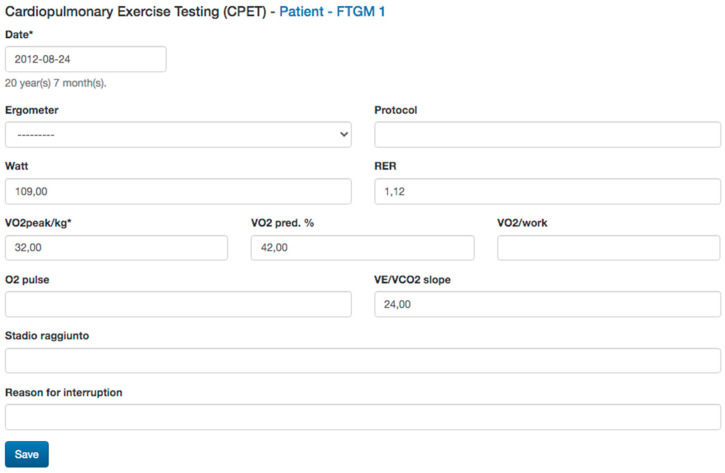
Cardiopulmonary exercise test data collected.

At the follow-up, any interventional or surgical procedures performed and symptoms or any events will be collected (Figure 6).

#### 2.5.3. Planned Analysis

The data collected would be suitable for longitudinal studies and outcome studies.

Any investigator can suggest a study by submitting the idea, starting with a hypothesis. The idea will be discussed among members and with the proponent/s. If the scientific committee accepts the idea, the dataset of the deidentified data needed for the study will be available to the proponent.

The mains endpoints collected in this clinical database will be:Sudden cardiac death;ICD implantation;Sustained ventricular arrhythmias;Non sustained ventricular arrhythmias;Atrial flutter, atrial fibrillation;Supraventricular tachycardia (SVT) consisted of an abrupt salve of three or more consecutive atrial premature beats at a rate of >100 beats per minute;Palpitations associated with syncope or near syncope in patients subsequently found to have inducible sustained SVT;Worsening of CMR data in particular dilation of the right ventricle, worsening of bi-ventricular function and heart failure.

## 3. Preliminary Results

Demographic, clinical, CPET, TTE, and CMR data are summarized in Table 1.

During the last years, the data of 880 repaired TOF (age at enrolment: 23.5 ± 12 years, range 10, 70 years) have been collected. The most frequent diagnosis was tetralogy of Fallot *n* = 797 (90%), while DORV Fallot type was *n* = 22 (3%). Age at surgical repair 1.71 (0.75–4.8) and trans-annular patch (TAP) was the most utilized technique for primary RVOT repair, *n* = 553 (63%).

The main characteristics of preliminary study population are listed in Table 2.

## 4. Discussion

TOF is the most frequent cyanotic CHD during childhood. In addition, TOF comprises the largest numbers of patients, repaired during their childhood, that currently have survived to adulthood. It has been established that TOF may present in both children and adults with a wide heterogeneity of anatomical, hemodynamical, and thoracic features that can be difficult to correct and address at the moment of their first cardiac surgery. These post-operative issues can also be challenging to manage, both medically or procedurally, during the early post-operative period and later on. In addition, this heterogeneity mentioned above may affect and determine later outcomes. In fact, it has been largely recognized that preoperative decisions determine not only the peri-operative outcomes but impacts also on the long-term complications of this population [13,16,24]. The extensive ventriculotomy and infundibulectomy, as well as the generous transannular patching of the RV outflow tract, which were routine surgical practice for TOF repair in the past, have been implicated in right ventricular dysfunction, as well as the genesis of fatal ventricular arrhythmias in this population. Thus, changing the approach to performing transannular patch repair (from transventricular to transatrial) has resulted in significant benefits [25].

During the follow-up, the principal aims of future surgical and procedural treatments of TOF patients is mainly conditioned by cardiac anatomy, secondly by its function beyond the patient’s clinical state and/or symptoms, and thirdly if TOF is accompanied by syndromes like Trisomy 21, RASopathies, or similar [24,25,26].

In this context, cardiac imaging plays a key role, especially CMR. However, cardiac dimensions, volumes, and velocities derive by the assumption that TOF hearts can be assimilated to “normal” ones; therefore, RV imaging parameters, achieved from the general population, are adapted. Along with the RV morphology and function, the pulmonary arterial pathway is also particularly important in TOF patients. The pulmonary trunk, main pulmonary arteries (right and left), and peripheral vessels may recognize differences if compared to normal subjects. Lastly, the tricuspid valve dysfunction can co-exist beside an inevitably right atrial wall surgical incision. This is particularly important when evaluating the entire right cardiac structures, which will never act as “normal” for all the reasons mentioned above. Furthermore, it must be considered that although the three-dimensional evaluation with CMR is the most accurate tool to evaluate the right ventricle in TOF patients, there is an interindividual and between individual’s variability in the evaluation of the right ventricle when using this method [10,22,27].

As a matter of fact, and despite an ongoing interest in the repaired TOF population, the clinical management of these patients and their risk stratification is still imperfect, needing to be refined and tailored to any single TOF patient. In this context, the implementation of standardized protocols (i.e., tissue characterization with delayed gadolinium enhancement and T1 mapping of the RV) seems to be necessary, even if not mandatory. Moreover, it has been recognized that the experience of any single institution is limited in this context. Different centers adopt different strategies to follow up, treat, and manage childhood repaired TOF during their adulthood [14].

We thus hypothesized that this large multicenter study would help to improve the risk stratification in this population. Furthermore, we hope and expect that the studies, implemented by the registry, will allow for well-powered outcomes and longitudinal studies.

Therefore, our aim with this large multicenter study should help to improve the risk stratification in this population. In fact, the right time to perform PVR in repaired TOF patients is still unclear, mainly in those who are asymptomatic. Thus far, basing PVR decision on MRI-assessed right ventricular size in these patients has failed. We found ourselves a population needing to replace the pulmonary prosthetic valve after a few years with a right ventricular re-dilatation associated with RV dysfunction. We must also consider that a percentage of these patients are younger than 18 years old. All this in the belief that the success in terms of survival of PVR was due to the normal return of the dimensions of the RV as well as the fact that the risk of sudden death at a certain threshold value of right ventricular dimension was higher than that of undergoing PVR with all the medium and long-term consequences of this approach. Only recently have some authors questioned these beliefs, and attempts are being made to postpone the timing of PVR in some completely asymptomatic patients without significant right ventricular dysfunction. Therefore, we hope that this registry will provide us with the possibility of identifying which patients with tetralogy of Fallot are, indeed, at risk of fatal ventricular arrhythmia and/or RV dysfunction and consequently to optimize the timing of PVR [23,24,25,26,27].

In this context, we prospect more and more collaboration between centers, and this should be mandatory in order to gain farther comprehension regarding possible and/or late complications of repaired TOF and to have enough understanding of these problems. In addition, the implementation of a national registry may permit to conduct further meaningful outcomes analyses, without the occurrence of single-center-driven bias management.

## 5. Conclusions

The multicenter repaired TOF clinical database will collect data of patients evaluated by CMR in most CHD centers of Italy. The registry has been set up to allow future research studies in this population to improve clinical/surgical management and risk stratification of this population.

## 6. Future Development

Potentials for future development are:Generating a structured report would allow standardization of the CMR report among the participating centers and could help centers with less expertise;Integration of the multicenter study Fallot study platform with PACS (picture archiving and communication system) through a specific customized intermediate software would allow for the delivery of anonymized DICOM images into a temporary storage;Extending the study to other Italian or Europeans centers.

## Figures and Tables

**Figure 1 diagnostics-13-03255-f001:**
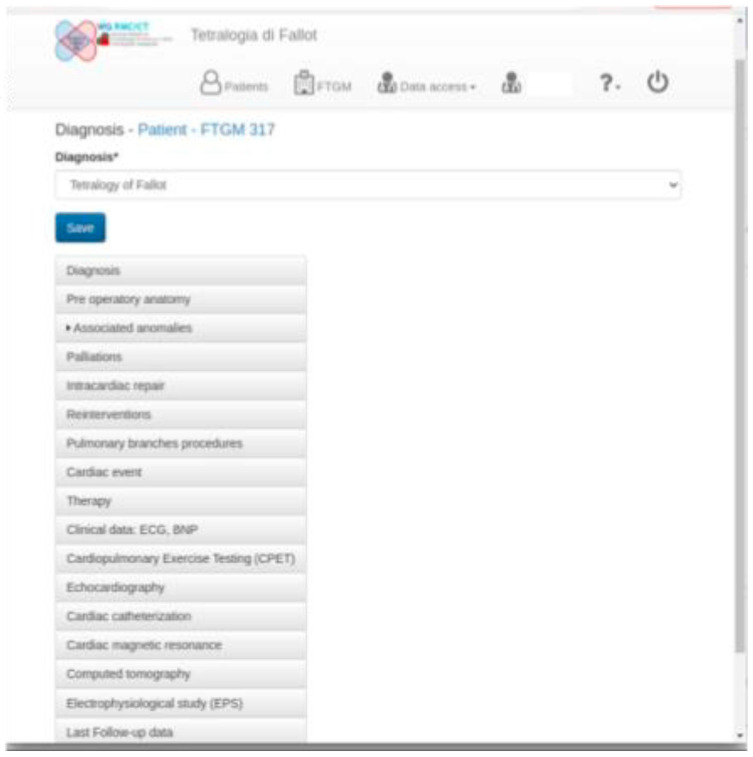
Home page of a patient compiled in the Fallot multicenter study website.

**Figure 2 diagnostics-13-03255-f002:**
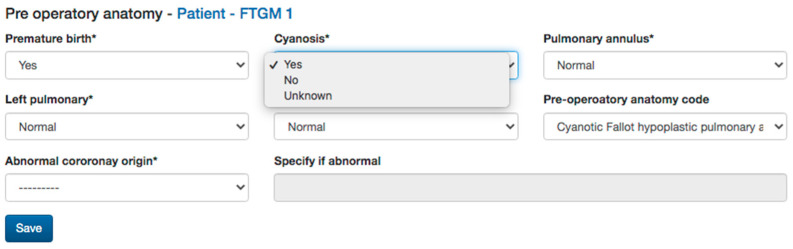
Pre-operative anatomy data collected in the Fallot multicenter study website.

**Figure 6 diagnostics-13-03255-f006:**
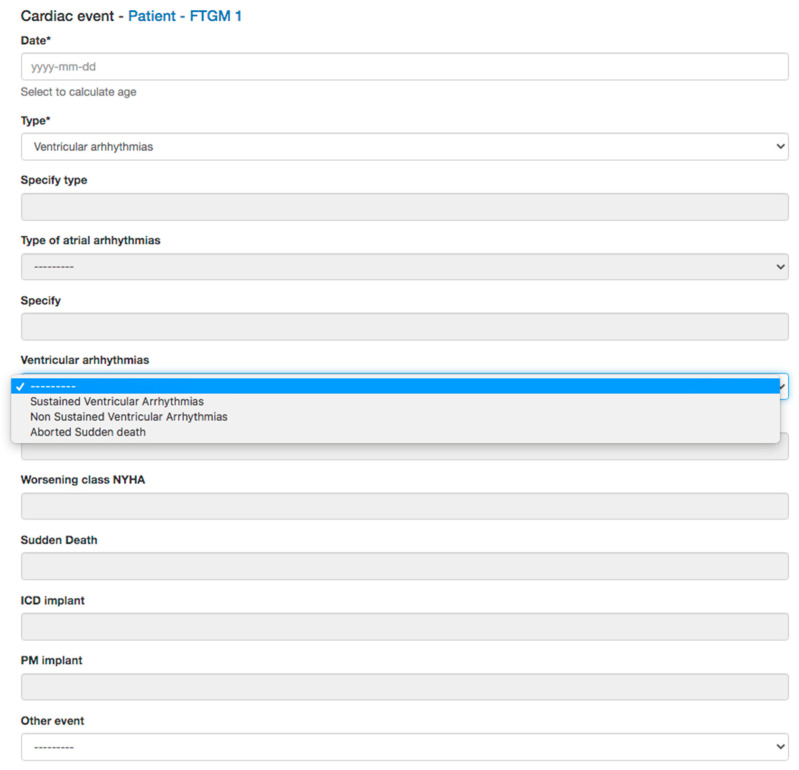
Adverse cardiac event data collected.

**Table 1 diagnostics-13-03255-t001:** Study population demographic and surgical characteristics.

Patients (Total)	N = 880
Tetralogy of Fallot (n, %)	797 (90%)
DORV Fallot type * (n, %)	22 (3%)
PA + VSD (n, %)	61 (7%)
Age at the last CMR (years) (mean ± SD)	23.5 ± 12
Previous shunt palliation (n, %)	288 (33%)
Age at primary repair (years)	1.71 (0.75–4.8)
**Type of primary RVOT repair**	
TAP (n, %)	553 (63%)
Infundibular patch/commissurotomy (n, %)	203 (23%)
Valved conduit/homograft (n, %)	75 (8.5%)
Unknown (n, %)	47 (5.3%)
Re-operated patients * (n, %)	357 (41%)

Legend: DORV: double outlet right ventricle; PA: pulmonary atresia; RVOT: right ventricular outflow tract; TAP: trans-annular patch; VSD: ventricular septal defect. SD: standard deviation. * Pulmonary valve implant (surgical or interventional) at the last CMR evaluation (many patients have CMR pre-reoperation).

**Table 2 diagnostics-13-03255-t002:** Study population preliminary data summary.

NT-Pro-BNP (*n* = 305) (Median, Q1, Q3)	11 (55, 195)
VO2/Kg/min (*n* = 350) (mean ± SD)	23.5 ± 7.4
QRS duration (ms) (*n*: 730) (mean ± SD)	140 ± 28
RVP (mmHg) (*n* = 391) (mean ± SD)	46 ± 19
Moderate/severe TR (%)	12%
LASVi (mL/m^2^) (*n* = 250) (mean ± SD)	32.5 ± 17
LVEDVi (mL/m^2^) (mean ± SD)	82.8 ± 16
LVEF (%) (mean ± SD)	58.3 ± 6.7
RASVi (mL/m^2^) (*n* = 280) (mean ± SD)	55 ± 23
RVEDVi (mL/m^2^) (mean ± SD)	137 ± 39
RVEF (%) (mean ± SD)	52 ± 7.7

**Legend:** LASVi: left atrium systolic volume indexed, LVEF: left ventricle ejection fraction, LVEDVi: left ventricle volume end-diastolic volume indexed; RASVi: right atrium systolic volume indexed, RVP: right ventricular pressure; RVEF: right ventricle ejection fraction, RVESVi: right ventricle volume end-systolic volume indexed, RVEDVi: right ventricle volume end-diastolic volume index; TR: tricuspid regurgitation.

## Data Availability

Data available on request due to privacy/ethical restrictions.

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
