# Peer review of "Overcoming Underpowering in the Outcome Analysis of Repaired—Tetralogy of Fallot: A Multicenter Database from the CMR/CT Working Group of the Italian Pediatric Cardiology Society (SICPed)"

_diagnostics, 2023, doi:10.3390/diagnostics13203255_

Round 1

Reviewer 1 Report

As suggested in the manuscript, the registry has been set up to allow future research studies in this CHD population to improve clinical/surgical management and risk stratification of this CHD population. Hence, this manuscript is clinically significant and interesting.

The quality of Figure 1 should be improved.

In the Introduction, the major damages of TOF should be introduced, as indicated by the following references.

[1]     Karali K, Makedou K, Kallifatidis A, Didagelos M, Giannakoulas G, Davos CH, Karamitsos TD, Ziakas A, Karvounis H, Hadjimiltiades S. The Interplay between Myocardial Fibrosis, Strain Imaging and Collagen Biomarkers in Adults with Repaired Tetralogy of Fallot. Diagnostics. 2021; 11(11): 2101.

[2]     Havers-Borgersen E, Butt JH, Smerup M, Gislason GH, Torp-Pedersen C, Gröning M, Schmidt MR, Søndergaard L, Køber L, Fosbøl EL. Incidence of Infective Endocarditis Among Patients With Tetralogy of Fallot. J Am Heart Assoc. 2021; 10(22): e022445.

[3]     Hosseinpour AR, González-Calle A, Adsuar-Gómez A, Ho SY. The Predicament of Surgical Correction of Tetralogy of Fallot. Pediatr Cardiol. 2021; 42(6): 1252-1257.

The quality of English language is good.

Author Response

Thank you very much for your comments and edits.

We have been trying to improve the quality of Figure 1 as suggested; since it is from the website platform, the latest one has the best quality that we have obtained.

We also updated the texts as suggested and introduced the references proposed. 

Reviewer 2 Report

The protocol study for the Italian Multicenter Registry of Repaired Tetralogy of Fallot (ToF) patients by Ait-Ali et al. wishes to address some essential issues in ToF management, proposing an accurate and simple method for overcoming underpowering in the outcome analysis of patients evaluated by CMR in many CHD centers in Italy. The Registry has also been set up to allow future research studies in this population to improve clinical/surgical management and risk stratification of this population.   The protocol and preliminary results are well-written and clear, and the findings suggest this Registry is a valid tool for better understanding this specific population's outcomes. In the "Preliminary results" session, the first line: "This section may be divided by subheadings..." is perhaps a typo that should be corrected accordingly. Please, check the whole manuscript for other typos and correct them appropriately.   Some minor comments: 1. Can you comment on how CMR could be temporarily applied in clinical practice in ToF?  2. Can you comment on the follow-up evaluation by different imaging techniques? 3. Can you comment on the relationship between the type of repair and clinical outcomes? 

Author Response

Thank you very much for your comments and edits. We really appreciated it.

As suggested, we corrected the "Preliminary results" session and re-checked the whole manuscript for typos. We also expanded the introduction and the discussion following the minor comments raised and updated the references accordingly.